# Adversarially Robust Federated Learning for Neural Networks

## Abstract

In federated learning, data is distributed among local clients which collaboratively train a prediction model using secure aggregation. To preserve the privacy of the clients, the federated learning paradigm requires each client to maintain a private local training data set, and only uploads its summarized model updates to the server. In this work, we show that this paradigm could lead to a vulnerable model, which collapses in performance when the corrupted data samples (under adversarial manipulations) are used for prediction after model deployment. To improve model robustness, we first decompose the aggregation error of the central server into bias and variance, and then, propose a robust federated learning framework, named Fed_BVA, that performs on-device adversarial training using the bias-variance oriented adversarial examples supplied by the server via asymmetrical communications. The experiments are conducted on multiple benchmark data sets using several prevalent neural network models, and the empirical results show that our framework is robust against white-box and black-box adversarial corruptions under both IID and non-IID settings.

## 1 Introduction

The explosive amount of decentralized user data collected from the ever-growing usage of smart devices, e.g., smartphones, wearable devices, home sensors, etc., has led to a surge of interest in the field of decentralized learning. To protect the privacy-sensitive data of the clients, federated learning (McMahan et al., 2017; Yang et al., 2019) has been proposed. Federated learning only allows a group of clients to train local models using their own data, and then collectively merges the model updates on a central server using secure aggregation (Acar et al., 2018). Due to its high privacy-preserving property, federated learning has attracted much attention in recent years along with the prevalence of efficient light-weight deep models (Howard et al., 2017) and low-cost network communications (Wen et al., 2017; Konečný et al., 2016).

In federated learning, the central server only inspects the secure aggregation of the local models as a whole. Consequently, it is susceptible to clients' corrupted updates (e.g., system failures, etc). Recently, multiple robust federated learning models (Fang et al., 2019; Pillutla et al., 2019; Portnoy & Hendler, 2020; Mostafa, 2019) have been proposed. These works only focus on performing client-level robust training or designing server-level aggregation variants with hyper-parameter tuning for Byzantine failures. However, none of them have the ability to mitigate the federated learning's vulnerability when the adversarial manipulations are present during testing, which as we shown in Section 4.1 that is mainly due to the generalization error in the model aggregation.

Our work bridges this gap by investigating the error incurred during the aggregation of federated learning from the perspective of bias-variance decomposition (Domingos, 2000; Valentini & Dietterich, 2004). Specifically, we show that the generalization error of the aggregated model on the central server can be decomposed as the combination of **bias** (triggered by the main prediction of these clients) and **variance** (triggered by the variations among clients' predictions). Next, we propose to perform the local robust training on clients by supplying them with a tiny amount of the bias-variance perturbed examples generated from the central server via asymmetrical communications. The experiments are conducted on neural networks with cross-entropy loss, however, other loss functions are also applicable as long as their gradients w.r.t. bias and variance are tractable to estimate. In this way, any gradient-based adversarial training strategies (Goodfellow et al., 2015; Madry et al., 2018) could be used. Compared with previous work, our major contributions include:

- We provide the exact solution of bias-variance analysis w.r.t. the generalization error which is perfectly suitable for neural network based federated learning. As a comparison, performing adversarial attacks or training with conventional federated learning methods will only focus on the bias of the central model but ignore the variance.

- We demonstrate that the conventional federated learning framework is vulnerable to the strong attacking methods with increasing communication rounds even if the adversarial training using the locally generated adversarial examples is performed on each client.
- Without violating the clients' privacy, we show that providing a tiny amount of bias-variance perturbed data from the central server to the clients through asymmetrical communication could dramatically improve the robustness of the training model under various settings.

## 2 PRELIMINARIES

### 2.1 SETTINGS

In federated learning, there is a central server and $K$ different clients, each with access to a private training set $\mathcal{D}_k = \{(x_i^k, t_i^k)\}_{i=1}^{n_k}$, where $x_i^k$, $t_i^k$, and $n_k$ are the features, label, and number of training examples in the $k^{\text{th}}$ client ($k = 1, \cdots, K$). Each data $\mathcal{D}_k$ is exclusively owned by client $k$ and will not be shared with the central server or other clients. In addition, there is a small public training set $\mathcal{D}_s = \{(x_j^s, t_j^s)\}_{j=1}^{n_s}$ with $n_s$ training examples from the server that is shared with clients, where $n_s \ll \sum_{k=1}^{K} n_k$. Note that this will not break the privacy constraints, for example, hospitals (local devices) that contribute to a federated learned medical image diagnosis system could take a few publicly accessible images as additional inputs. The goal of federated learning is to train a global classifier $f(\cdot)$ using knowledge from all the clients such that it generalizes well over test data $\mathcal{D}_{test}$. The notation used in this paper is summarized in the Appendix (see Table 4).

### 2.2 PROBLEM DEFINITION

In this paper, we study the adversarial robustness of neural networks[1] in federated learning setting, and we define robust decentralized learning as follows.

**Definition 2.1. (Adversarially Robust Federated Learning)**
***Input:*** (1) A set of private training data $\{\mathcal{D}_k\}_{k=1}^{K}$ on $K$ different clients; (2) Tiny amount of training data $\mathcal{D}_s$ on the central server; (3) Learning algorithm $f(\cdot)$ and loss function $L(\cdot, \cdot)$.
***Output:*** A trained model on the central server that is robust against adversarial perturbation.

We would like to point out that our problem definition has the following properties: **Asymmetrical communication:** The asymmetrical communication between each client and server cloud is allowed: the server provides both global model parameters and limited shared data to the clients; while each client only uploads its local model parameters back to the server. **Data distribution:** All training examples on the clients and the server are assumed to follow the same data distribution. However, the experiments show that our proposed algorithm also achieves outstanding performance under the non-IID setting, which could be common among personalized clients in real scenarios. **Shared learning algorithm:** All the clients are assumed to use the identical model $f(\cdot)$, including architectures as well as hyper-parameters (e.g., learning rate, local epochs, local batch size).

**Remark.** *The basic assumption of this problem setting is that the learning process is clean (no malicious behaviors are observed during training), however, the intentionally generated adversarial poisoning data will be mixed with clean data during training. The eventual trained model being deployed on the devices will be robust against potential future adversarial attacks.*

### 2.3 BIAS-VARIANCE TRADE-OFF

Following (Domingos, 2000; Valentini & Dietterich, 2004), we define the optimal prediction, main prediction as well as the bias, variance, and noise for any real-valued loss function $L(\cdot, \cdot)$ as follows:

**Definition 2.2. (Optimal Prediction and Main Prediction)** Given loss function $L(\cdot, \cdot)$ and learning algorithm $f(\cdot)$, optimal prediction $y_*$ and main prediction $y_m$ for an example are defined as:
$$y_*(x) = \arg\min_{y} \mathbb{E}_t[L(y, t)] \quad \text{and} \quad y_m(x) = \arg\min_{y'} \mathbb{E}_\mathcal{D}[L(f_\mathcal{D}(x), y')] \qquad (1)$$
where $t$ and $\mathcal{D}$ are viewed as the random variables to denote the class label and training set, and $f_\mathcal{D}$ denotes the model trained on $\mathcal{D}$. In short, the main prediction is the prediction whose average loss relative to all the predictions over data distributions is minimum, e.g., the main prediction for zero-one loss is the mode of predictions. In this work, we show that the main prediction is the average prediction of client models for mean squared (MSE) loss and cross-entropy (CE) loss in Section 4.1.

---

[1] Our theoretical contribution mainly focuses on classification using neural networks with cross-entropy loss and mean squared loss. However, the proposed framework is generic to allow the use of other classification loss functions as well.

**Definition 2.3.** (**Bias, Variance and Noise**) Given a loss function $L(\cdot, \cdot)$ and a learning algorithm $f(\cdot)$, the expected loss $\mathbb{E}_{\mathcal{D},t}[L(f_{\mathcal{D}}(x), t)]$ for an example $x$ can be decomposed[2] into bias, variance and noise as follows:

$$B(x) = L(y_m, y_*) \quad \text{and} \quad V(x) = \mathbb{E}_{\mathcal{D}}[L(f_{\mathcal{D}}(x), y_m)] \quad \text{and} \quad N(x) = \mathbb{E}_t[L(y_*, t)] \quad (2)$$

In short, bias is the loss incurred by the main prediction w.r.t. the optimal prediction, and variance is the average loss incurred by predictions w.r.t. the main prediction. Noise is conventionally assumed to be irreducible and independent to $f(\cdot)$.

**Remark.** *Our definitions on optimal prediction, main prediction, bias, variance and noise slightly differ from previous ones (Domingos, 2000; Valentini & Dietterich, 2004). For example, conventional optimal prediction was defined as $y_*(x) = \arg\min_y \mathbb{E}_t[L(t, y)]$, and it is equivalent to our definition when loss function is symmetric over its arguments, i.e., $L(y_1, y_2) = L(y_2, y_1)$. Note that this decomposition holds for any real-valued loss function in the binary setting (Domingos, 2000) with a bias & variance trade-off coefficient that has a closed-form expression. For multi-class setting, we inherit their definition of bias & variance directly, and treat the trade-off coefficient as a hyper-parameter $\lambda$ to tune because no closed-form expression of $\lambda$ is available.*

## 3 THE PROPOSED FRAMEWORK

A typical framework (Kairouz et al., 2019) of privacy-preserving federated learning can be summarized as follows: (1) *Client Update:* Each client updates local model parameters $w_k$ by minimizing the empirical loss over its own training set; (2) *Forward Communication:* Each client uploads its model parameter update to the central server; (3) *Server Update:* It synchronously aggregates the received parameters; (4) *Backward Communication:* The global parameters are sent back to the clients. Our framework follows the same paradigm but with substantial modifications as below.

**Server Update**. The server has two components: The first one uses FedAvg (McMahan et al., 2017) algorithm to aggregate the local models' parameters, i.e., $w_G = \text{Aggregate}(w_1, \cdots, w_K) = \sum_{k=1}^{K} \frac{n_k}{n} w_k$ where $n = \sum_{k=1}^{K} n_k$ and $w_k$ is the model parameters in the $k^{\text{th}}$ client. Meanwhile, another component is designed to produce adversarially perturbed examples which could be induced by a poisoning attack algorithm for the usage of robust adversarial training.

It has been well studied (Belkin et al., 2019; Domingos, 2000; Valentini & Dietterich, 2004) that in the classification setting, the generalization error of a learning algorithm on an example is determined by the bias, variance, and irreducible noise as defined in Eq. (2). Similar to the previous work, we also assume a noise-free learning scenario where the class label $t$ is a deterministic function of $x$ (i.e., if $x$ is sampled repeatedly, the same values of its class $t$ will be observed). This motivates us to generate the adversarial examples by attacking the bias and variance induced by clients' models as:

$$\max_{\hat{x} \in \Omega(x)} B(\hat{x}; w_1, \cdots, w_K) + \lambda V(\hat{x}; w_1, \cdots, w_K) \quad \forall (x, t) \in \mathcal{D}_s \quad (3)$$

where $B(\hat{x}; w_1, \cdots, w_K)$ and $V(\hat{x}; w_1, \cdots, w_K)$ could be empirically estimated from a finite number of clients' parameters trained on local training sets $\{\mathcal{D}_1, \mathcal{D}_2, \cdots, \mathcal{D}_K\}$. Here $\lambda$ is a hyper-parameter to measure the trade-off of bias and variance, and $\Omega(x)$ is the perturbation constraint.

Note that $\mathcal{D}_s$ (on the server) is the candidate subset of all available training examples that would lead to their perturbed counterparts. This is a more feasible setting as compared to generating adversarial examples on clients' devices because the server usually has much powerful computational capacity in real scenarios that allows the usage of flexible poisoning attack algorithms. In this case, both poisoned examples and server model parameters would be sent back to each client (*Backward Communication*), while only clients' local parameters would be uploaded to the server (*Forward Communication*), i.e., the *asymmetrical communication* as discussed in Section 2.2.

**Client Update**. The robust training of one client's prediction model (i.e., $w_k$) can be formulated as the following minimization problem.

$$\min_{w_k} \left( \sum_{i=1}^{n_k} L(f_{\mathcal{D}_k}(x_i^k; w_k), t_i^k) + \sum_{j=1}^{n_s} L(f_{\mathcal{D}_k}(\hat{x}_j^s; w_k), t_j^s) \right) \quad (4)$$

where $\hat{x}_j^s \in \Omega(x_j^s)$ is the perturbed examples that is asymmetrically transmitted from the server.

---

[2] This decomposition is based on the weighted sum of bias, variance, and noise. In general, $t$ is a non-deterministic function (Domingos, 2000) of $x$ when the irreducible noise is considered. Namely, if $x$ is sampled repeatedly, different values of $t$ will be observed.

**Remark.** *Intuitively, the bias measures the systematic loss of a learning algorithm, and the variance measures the prediction consistency of the learner over different training sets. Therefore, our robust federated learning framework has the following advantages: (i) it encourages the clients to consistently produce the optimal prediction for perturbed examples, thereby leading to a better generalization performance; (ii) local adversarial training on perturbed examples allows to learn a robust local model, and thus a robust global model could be aggregated from clients.*

Theoretically, we could still have another alternative robust federated training strategy:

$$\min_{w_k} \sum_{i=1}^{n_k} \max_{\hat{x}_i^k \in \Omega(x_i^k)} L(f(\hat{x}_i^k; w_k), t_i^k) \quad \forall k \in \{1, 2, \cdots, K\} \tag{5}$$

where the perturbed training examples of each client $k$ is generated on local devices from $\mathcal{D}_k$ instead of transmitted from the server. This min-max formula is similar to (Madry et al., 2018; Tramèr et al., 2018) where the inner maximization problem synthesizes the adversarial counterparts of clean examples, while the outer minimization problem finds the optimal model parameters over perturbed training examples. Thus, each local robust model is trained individually, nevertheless, poisoning attacks on device will largely increase the computational cost and memory usage. Meanwhile, it only considers the client-specific loss and is still vulnerable against adversarial examples with increasing communication rounds. Both phenomena are observed in our experiments (see Fig. 4 and Fig. 5).

## 4 ALGORITHM

### 4.1 BIAS-VARIANCE ATTACK

We first consider the maximization problem in Eq. (3) using bias-variance based adversarial attacks. It aims to find the adversarial example $\hat{x}$ (from the original example $x$) that would produce large bias and variance values w.r.t. clients' local models. Specifically, perturbation constraint $\hat{x} \in \Omega(x)$ forces the adversarial example $\hat{x}$ to be visually indistinguishable w.r.t. $x$. Here we consider the well-studied $l_\infty$-bounded adversaries[3] (Goodfellow et al., 2015; Madry et al., 2018; Tramèr et al., 2018) such that $\Omega(x) := \{\hat{x} | ||\hat{x} - x||_\infty \le \epsilon\}$ for a perturbation magnitude $\epsilon$. Furthermore, we propose to consider the following two gradient-based algorithms to generate adversarial examples.

**Bias-variance based Fast Gradient Sign Method (BV-FGSM):** Following FGSM (Goodfellow et al., 2015), it linearizes the maximization problem in Eq. (3) with one-step attack as follows.

$$\hat{x}_{\text{BV-FGSM}} := x + \epsilon \cdot \text{sign}\left(\nabla_x \left(B(x; w_1, \cdots, w_K) + \lambda V(x; w_1, \cdots, w_K)\right)\right) \tag{6}$$

**Bias-variance based Projected Gradient Descent (BV-PGD):** PGD can be considered as a multi-step variant of FGSM (Kurakin et al., 2017) and might generate powerful adversarial examples. This motivated us to derive a BV-based PGD attack:

$$\hat{x}_{\text{BV-PGD}}^{l+1} := \text{Proj}_{\Omega(x)}\left(\hat{x}^l + \epsilon \cdot \text{sign}\left(\nabla_{\hat{x}^l}\left(B(\hat{x}^l; w_1, \cdots, w_K) + \lambda V(\hat{x}^l; w_1, \cdots, w_K)\right)\right)\right) \tag{7}$$

where $\hat{x}^l$ is the adversarial example at the $l^{\text{th}}$ step with the initialization $\hat{x}^0 = x$ and $\text{Proj}_{\Omega(x)}(\cdot)$ projects each step onto $\Omega(x)$.

**Remark.** *The proposed framework could be naturally generalized to any gradient-based adversarial attack algorithms where the gradients of bias $B(\cdot)$ and variance $V(\cdot)$ w.r.t. $x$ are tractable when estimated from finite training sets. Compared with the existing attack methods (Carlini & Wagner, 2017; Goodfellow et al., 2015; Kurakin et al., 2017; Moosavi-Dezfooli et al., 2016), our loss function the adversary aims to optimize is a linear combination of bias and variance, whereas existing work mainly focused on attacking the overall classification error that considers bias only.*

The following theorem states that bias $B(\cdot)$ and variance $V(\cdot)$ as well as their gradients over input $x$ could be estimated using the clients' models.

**Theorem 4.1.** Assume that $L(\cdot, \cdot)$ is the cross-entropy loss function, then, the empirical estimated main prediction $y_m$ for an input example $(x, t)$ has the following closed-form expression: $y_m(x; w_1, \cdots, w_K) = \frac{1}{K} \sum_{k=1}^{K} f_{\mathcal{D}_k}(x; w_k)$. Furthermore, the empirical bias and variance, as well as their gradients over an input $x$ are estimated as follows:

$$B(x; w_1, \cdots, w_K) = \frac{1}{K} \sum_{k=1}^{K} L(f_{\mathcal{D}_k}(x; w_k), t); \quad V(x; w_1, \cdots, w_K) = L(y_m, y_m) = H(y_m)$$

---

[3]$l_\infty$ robustness is surely not the only option for robustness learning. However, we use this standard approach to show the limitations of prior federated learning, and evaluate the improvements of our proposed framework.

Here, $H(y_m) = -\sum_{j=1}^{C} y_m^{(j)} \log y_m^{(j)}$ is the entropy of the main prediction $y_m$ and $C$ is the number of classes. Easily, we can have their gradients in terms of the bias and variance as $\nabla_x B(x; w_1, \cdots, w_K) = \frac{1}{K} \sum_{k=1}^{K} \nabla_x L(f_{\mathcal{D}_k}(x; w_k), t)$ and $\nabla_x V(x; w_1, \cdots, w_K) = -\frac{1}{K} \sum_{k=1}^{K} \sum_{j=1}^{C} (\log y_m^{(j)} + 1) \nabla_x f_{\mathcal{D}_k}^{(j)}(x; w_k)$. Details of the proof is elaborated in A.2.

In addition, we also consider the case where $L(\cdot, \cdot)$ is the MSE loss function. But the gradients of MSE's bias and variance are much more computational demanding comparing with the concise formulas that cross-entropy ends up with. More comparisons are illustrated in Appendix A.5.1.

---

**Algorithm 1** Fed_BVA

1: **Input:** $K$ (number of clients, with local data sets $\{\mathcal{D}_k\}_{k=1}^{K}$); $f$ (learning model), $E$ (number of local epochs); $F$ (fraction of clients selected on each round); $B$ (batch size of local client); $\eta$ (learning rate); $\mathcal{D}_s$ (shared data set on server); $\epsilon$ (perturbation magnitude).
2: **Initialization:** Initialize $w_G^0$ and $\hat{\mathcal{D}}_s = \emptyset$
3: **for** each round $r = 1, 2, \cdots$ **do**
4: $\quad m = \max(F \cdot K, 1)$
5: $\quad S_r \leftarrow$ randomly sampled $m$ clients
6: $\quad$ **for** each client $k \in S_r$ in parallel **do**
7: $\quad\quad w_k^r, f_{\mathcal{D}_k}, \nabla_x f_{\mathcal{D}_k} \leftarrow$
$\quad\quad$ **ClientUpdate**$(w_G^{r-1}, \hat{\mathcal{D}}_s, \mathcal{D}_s, k)$
8: $\quad$ **end for**
9: $\quad \hat{\mathcal{D}}_s \leftarrow$ **BVAttack**$(\{f_{\mathcal{D}_k}, \nabla_x f_{\mathcal{D}_k}\} | k \in S_r)$
10: $\quad w_G^r \leftarrow$ **Aggregate**$(w_k^r | k \in S_r)$
11: **end for**
12: **return** $w_G$

---

**Algorithm 2** ClientUpdate$(w, \hat{\mathcal{D}}_s, \mathcal{D}_s, k)$

1: Initialize $k^{\text{th}}$ client's model with $w$
2: $\mathcal{B} \leftarrow$ split $\mathcal{D}_k \cup \hat{\mathcal{D}}_s$ into batches of size $B$
3: **for** each local epoch $i = 1, 2, \cdots, E$ **do**
4: $\quad$ **for** local batch $(x, t) \in \mathcal{B}$ **do**
5: $\quad\quad w \leftarrow w - \eta \nabla L(f_{\mathcal{D}_k}(x; w), t)$
6: $\quad$ **end for**
7: **end for**
8: Calculate $f_{\mathcal{D}_k}(x; w_k^r), \nabla_x f_{\mathcal{D}_k}(x; w) \, \forall x \in \mathcal{D}_s$
9: **return** $w, f_{\mathcal{D}_k}(x; w_k^r), \nabla_x f_{\mathcal{D}_k}(x; w)$

---

**Algorithm 3** BVAttack$(\{f_{\mathcal{D}_k}, \nabla_x f_{\mathcal{D}_k}\} | k \in S_r)$

1: Initialize $\hat{\mathcal{D}}_s = \emptyset$
2: **for** $(x, t) \in \mathcal{D}_s$ **do**
3: $\quad$ Estimate the gradients $\nabla_x B(x)$ and $\nabla_x V(x)$ using Theorem 4.1
4: $\quad$ Calculate $\hat{x}$ using Eq. (6) or (7) and add to $\hat{\mathcal{D}}_s$
5: **end for**
6: **return** $\hat{\mathcal{D}}_s$

---

### 4.2 FED_BVA

We present a novel robust federated learning algorithm with our proposed bias-variance attacks, named Fed_BVA. Following the framework defined in Eq. (3) and Eq. (4), key components of our algorithm are (1) bias-variance attacks for generating adversarial examples on the server, and (2) adversarial training using poisoned server examples together with clean local examples on each client. Therefore, we optimize these two objectives by producing the adversarial examples $\hat{\mathcal{D}}_s$ and updating the local model parameters $w$ iteratively.

The proposed algorithm is summarized in Alg. 1. Given the server's $\mathcal{D}_s$ and clients' training data $\{\mathcal{D}_k\}_{k=1}^{K}$ as input, the output is a robust global model on the server. In this case, the clean server data $\mathcal{D}_s$ will be shared to all the clients. First, it initializes the server's model parameter $w_G$ and perturbed data $\hat{\mathcal{D}}_s$, and then assigns to the randomly selected clients (Steps 4-5). Next, each client optimizes its own local model (Steps 6-8) with the received global parameters $w_G$ as well as its own clean data $\mathcal{D}_k$, and uploads the updated parameters as well as the gradients of local model on each shared server example back to the server. At last, the server generates the perturbed data $\hat{\mathcal{D}}_s$ (Step 9) using the proposed bias-variance attack algorithm (see Alg. 3) with aggregations (model parameter average, bias gradients average, and variance gradients average) in the similar manner as FedAvg (McMahan et al., 2017). These aggregations can be privacy secured if additive homomorphic encryption (Acar et al., 2018) is applied.

## 5 EXPERIMENTS

### 5.1 SETTINGS

In this section, we evaluate the adversarial robustness of our proposed algorithm on four benchmark data sets: MNIST[4], Fashion-MNIST[5], CIFAR-10[6] and CIFAR-100[6]. The baseline models

---

[4] http://yann.lecun.com/exdb/mnist
[5] https://github.com/zalandoresearch/fashion-mnist
[6] https://www.cs.toronto.edu/~kriz/cifar.html

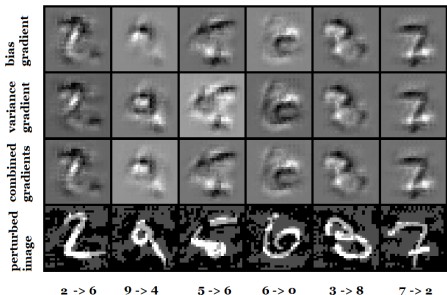

Figure 1: Visualizations of bias, variance, bias+variance, and perturbed images for MNIST.

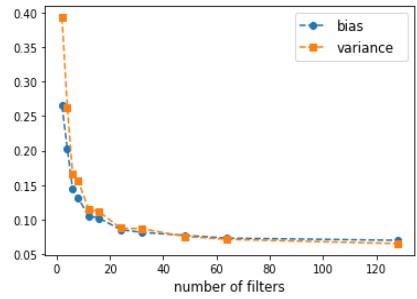

Figure 2: Bias-variance curve w.r.t. the CNN model complexity on MNIST.

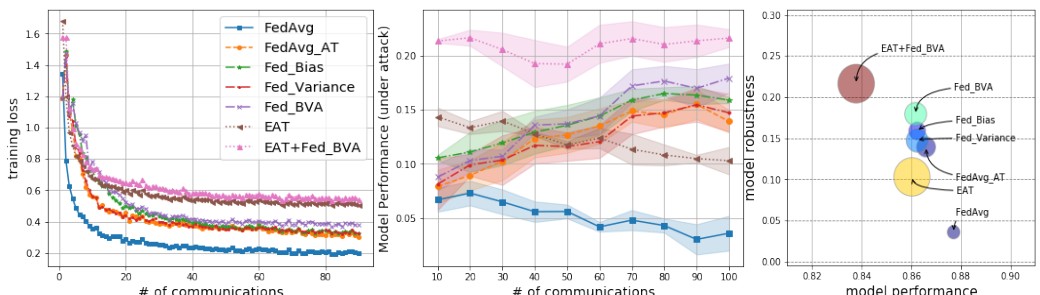

Figure 3: Convergence on Fashion-MNIST (PGD-20 attack)

Figure 4: Performance on Fashion-MNIST (PGD-20 attack)

Figure 5: Efficiency on Fashion-MNIST (PGD-20 attack)

we used include: (1). Centralized: the training with one centralized model, which is identical to the federated learning case that only has one client ($K = 1$) with fraction ($F = 1$). (2). FedAvg: the classical federated averaging model (McMahan et al., 2017). (3). FedAvg_AT: The simplified version of our proposed method where the local clients perform adversarial training with the asymmetrical transmitted perturbed data generated on top of FedAvg's aggregation. (4) - (6). Fed_Bias, Fed_Variance, Fed_BVA: Our proposed methods where the asymmetrical transmitted perturbed data is generated using the gradients of bias-only attack, variance-only attack, and bias-variance attack, respectively. (7). EAT: Ensemble adversarial training (Tramèr et al., 2018), where each client performs local adversarial training using Eq. (5), and their model updates are aggregated on server using FedAvg. For fair comparisons, all baselines are modified to the asymmetrical communications setting (FedAvg and EAT have clean $\mathcal{D}_s$ received), and all their initializations are set to be the same. (8). EAT+Fed_BVA: A combination of baselines (6) and (7). Note that baselines (7) and (8) have high computational requirements on client devices, and are usually not preferred in real scenarios.

For the defense model, we use a 4-layer CNN model for MNIST and Fashion-MNIST, and VGG9 architecture for CIFAR-10 and CIFAR-100. Regarding blackbox attacks, we apply ResNet18 (He et al., 2016), VGG11 (Simonyan & Zisserman, 2015), Xception (Chollet, 2017), and MobileNetV2 (Sandler et al., 2018) for CIFAR data, and provide a variety of models for MNIST and Fashion-MNIST by following the design of (Tramèr et al., 2018). The training is performed using the SGD optimizer with fixed learning rate of 0.01 and momentum of value 0.9. The trade-off coefficient between bias and variance is set to $\lambda = 0.01$ for all experiments. All hyper-parameters of federated learning are presented in Table 5 in the Appendix. We empirically demonstrate that these hyper-parameter settings are preferable in terms of both training accuracy and robustness (see the details of Fig. 6 - Fig. 14 in the Appendix). To evaluate the robustness of our federated learning algorithm against adversarial attacks, except for the clean model training, we perform FGSM (Goodfellow et al., 2015), PGD (Kurakin et al., 2017) with 10 and 20 steps towards the aggregated server model on the $\mathcal{D}_{test}$. Following (Tramèr et al., 2018; Wang et al., 2019), the maximum perturbations allowed are $\epsilon = 0.3$ on MNIST and Fashion-MNIST, and $\epsilon = \frac{16}{255}$ on CIFAR-10 and CIFAR-100 for both threat and defense models. For IID sampling, the data is shuffled and uniformly partitioned into each client; For non-IID setting, data is divided into $2F \cdot K$ shards based on sorted labels, then assigns each client with 2 shards. Thereby, each client will have data with at most two classes.

## 5.2 RESULT ANALYSIS

To analyze the properties of our proposed Fed_BVA framework, we present two visualization plots on MNIST using a trained CNN model where the bias and variance are both calculated on the training examples. In Fig. 1, we visualize the extracted gradients using adversarial attack from bias, variance, and bias-variance. Notice that the gradients of bias and variance are similar but with subtle differences in local pixel areas. However, according to Theorem 4.1, the gradient calculation of these two are quite different: bias requires the target label as input, but variance only needs the model output and main prediction. From another perspective, we also investigate the bias-variance magnitude relationship with varying model complexity. As shown in Fig. 2, with increasing model complexity (more convolutional filters in CNN), both bias and variance decrease. This result is different from the double-descent curve or bell-shape variance curve claimed in (Belkin et al., 2019; Yang et al., 2020). The reasons are twofold: First, their bias-variance definitions are from the MSE regression decomposition perspective, whereas our decomposition utilizes the concept of main prediction, and the generalization error is decomposed from the classification perspective; Second, their implementations only evaluate the bias and variance using training batches on one central model and thus is different from the definition which requires the variance to be estimated from multiple sub-models (in our scenario, client models).

The convergence plot of all baselines is presented in Fig. 3. We observe that FedAvg has the best convergence, and all robust training will have a slightly higher loss upon convergence. This matches the observations in (Madry et al., 2018) which state that training performance may be sacrificed in order to provide robustness for small capacity networks. For the model performance shown in Fig. 4, we observe that the aggregation of federated learning is vulnerable to adversarial attacks since both FedAvg and EAT have decreased performance with an increasing number of server-client communications. Other baselines that utilized the asymmetrical communications have increasing robustness with more communication rounds although only a small number of perturbed examples ($n_s = 64$) are transmitted. We also observe that when communication rounds reach 40, Fed_BVA starts to outperform EAT while the latter is even more resource-demanding than Fed_BVA (shown in Fig. 5, where the pie plot size represents the running time). Overall, bias-variance based adversarial training via asymmetric communication is both effective and efficient for robust federated learning.

For the comprehensive experiments in Table 1 and Table 2, it is easy to verify that our proposed model outperforms all other baselines regardless of the source of the perturbed examples (i.e., locally generated like EAT+Fed_BVA or asymmetrically transmitted from the server like Fed_BVA). Comparing with standard robust federated learning FedAvg_AT, the performance of Fed_BVA against adversarial attacks still increases $4\% - 13\%$ and $2\% - 9\%$ on IID and non-IID settings respectively, although Fed_BVA is theoretically suitable for the cases that clients have IID samples. In Table 3, we observe a similar trend where Fed_BVA outperforms FedAvg_AT on CIFAR-10 and CIFAR-100 (with $0.2\% - 10\%$ increases) when defending different types of adversarial examples. Comparing with strong local adversarial training baseline EAT, we also observe a maximum 13% accuracy increase when applying its bias-variance oriented baseline EAT+Fed_BVA. Overall, the takeaway is that without local adversarial training, using a bias-variance based robust learning framework will almost always outperform other baselines for defending FGSM and PGD attacks. When local adversarial training is allowed (e.g., client device has powerful computation ability), using bias-variance robust learning with local adversarial training will mostly have the best robustness.

We also conducted various additional experiments in Appendix A.5 which includes: (1) Comparison of efficiency and effectiveness of Fed_BVA using cross-entropy loss and MSE loss; (2) Comparison of single-step Fed_BVA and multi-step Fed_BVA in terms of the generation of $\hat{\mathcal{D}}_s$; (3) Three training scenarios of Fed_BVA that use client-specific adversarial examples or universal adversarial examples; (4) Ablation study in terms of the number of shared perturb examples $n_s$, optimizer's momentum, and the number of local epochs $E$; (5) Blackbox attacking transferability between various models on all four data sets under multiple settings.

## 6 RELATED WORK

**Adversarial Machine Learning:** While machine learning models have achieved remarkable performance over clean inputs, recent work (Goodfellow et al., 2015) showed that those trained models are vulnerable to adversarially chosen examples by adding the imperceptive noise to the clean inputs. In general, the adversarial robustness of centralized machine learning models have been explored from the following aspects: adversarial attacks (Carlini & Wagner, 2017; Athalye et al., 2018; Zhu et al.,

| | IID | | | | non-IID | | | |
|---|---|---|---|---|---|---|---|---|
| | Clean | FGSM | PGD-10 | PGD-20 | Clean | FGSM | PGD-10 | PGD-20 |
| Centralized | **0.991**$_{\pm 0.000}$ | 0.689$_{\pm 0.000}$ | 0.260$_{\pm 0.000}$ | 0.182$_{\pm 0.000}$ | n/a | n/a | n/a | n/a |
| FedAvg | 0.989$_{\pm 0.001}$ | 0.669$_{\pm 0.009}$ | 0.576$_{\pm 0.005}$ | 0.267$_{\pm 0.014}$ | 0.980$_{\pm 0.002}$ | 0.491$_{\pm 0.067}$ | 0.475$_{\pm 0.057}$ | 0.158$_{\pm 0.074}$ |
| FedAvg_AT | 0.988$_{\pm 0.000}$ | 0.802$_{\pm 0.001}$ | 0.745$_{\pm 0.014}$ | 0.512$_{\pm 0.042}$ | 0.974$_{\pm 0.005}$ | 0.649$_{\pm 0.066}$ | 0.615$_{\pm 0.045}$ | 0.363$_{\pm 0.066}$ |
| Fed_Bias | 0.986$_{\pm 0.000}$ | 0.812$_{\pm 0.009}$ | 0.788$_{\pm 0.021}$ | 0.583$_{\pm 0.036}$ | 0.971$_{\pm 0.004}$ | 0.679$_{\pm 0.040}$ | 0.627$_{\pm 0.078}$ | 0.394$_{\pm 0.103}$ |
| Fed_Variance | 0.985$_{\pm 0.001}$ | 0.803$_{\pm 0.007}$ | 0.779$_{\pm 0.014}$ | 0.572$_{\pm 0.019}$ | 0.973$_{\pm 0.005}$ | 0.684$_{\pm 0.004}$ | 0.622$_{\pm 0.049}$ | 0.395$_{\pm 0.049}$ |
| Fed_BVA | 0.986$_{\pm 0.001}$ | 0.818$_{\pm 0.003}$ | 0.804$_{\pm 0.009}$ | 0.613$_{\pm 0.020}$ | 0.969$_{\pm 0.002}$ | 0.705$_{\pm 0.009}$ | 0.664$_{\pm 0.013}$ | 0.469$_{\pm 0.031}$ |
| EAT | 0.981$_{\pm 0.000}$ | **0.902**$_{\pm 0.001}$ | 0.907$_{\pm 0.001}$ | 0.811$_{\pm 0.004}$ | 0.972$_{\pm 0.002}$ | 0.789$_{\pm 0.016}$ | 0.721$_{\pm 0.018}$ | 0.415$_{\pm 0.035}$ |
| EAT+Fed_BVA | 0.980$_{\pm 0.001}$ | 0.901$_{\pm 0.006}$ | **0.910**$_{\pm 0.004}$ | **0.821**$_{\pm 0.013}$ | 0.965$_{\pm 0.005}$ | **0.811**$_{\pm 0.020}$ | **0.831**$_{\pm 0.013}$ | **0.670**$_{\pm 0.014}$ |

Table 1: Accuracy of MNIST under white-box attacks in IID and non-IID settings

| | IID | | | | non-IID | | | |
|---|---|---|---|---|---|---|---|---|
| | Clean | FGSM | PGD-10 | PGD-20 | Clean | FGSM | PGD-10 | PGD-20 |
| Centralized | **0.882**$_{\pm 0.000}$ | 0.229$_{\pm 0.000}$ | 0.010$_{\pm 0.000}$ | 0.009$_{\pm 0.000}$ | n/a | n/a | n/a | n/a |
| FedAvg | 0.877$_{\pm 0.001}$ | 0.300$_{\pm 0.021}$ | 0.072$_{\pm 0.016}$ | 0.036$_{\pm 0.016}$ | **0.804**$_{\pm 0.013}$ | 0.193$_{\pm 0.036}$ | 0.061$_{\pm 0.015}$ | 0.017$_{\pm 0.003}$ |
| FedAvg_AT | 0.866$_{\pm 0.001}$ | 0.490$_{\pm 0.021}$ | 0.170$_{\pm 0.014}$ | 0.139$_{\pm 0.011}$ | 0.730$_{\pm 0.023}$ | 0.445$_{\pm 0.065}$ | 0.136$_{\pm 0.044}$ | 0.087$_{\pm 0.042}$ |
| Fed_Bias | 0.862$_{\pm 0.001}$ | 0.505$_{\pm 0.015}$ | 0.199$_{\pm 0.007}$ | 0.159$_{\pm 0.003}$ | 0.709$_{\pm 0.025}$ | 0.460$_{\pm 0.038}$ | 0.149$_{\pm 0.067}$ | 0.115$_{\pm 0.054}$ |
| Fed_Variance | 0.862$_{\pm 0.002}$ | 0.496$_{\pm 0.012}$ | 0.201$_{\pm 0.012}$ | 0.157$_{\pm 0.017}$ | 0.719$_{\pm 0.036}$ | 0.499$_{\pm 0.081}$ | **0.188**$_{\pm 0.025}$ | **0.120**$_{\pm 0.038}$ |
| Fed_BVA | 0.862$_{\pm 0.003}$ | 0.528$_{\pm 0.016}$ | 0.210$_{\pm 0.023}$ | 0.180$_{\pm 0.027}$ | 0.710$_{\pm 0.045}$ | 0.495$_{\pm 0.030}$ | 0.141$_{\pm 0.021}$ | 0.093$_{\pm 0.028}$ |
| EAT | 0.860$_{\pm 0.005}$ | **0.773**$_{\pm 0.029}$ | 0.191$_{\pm 0.012}$ | 0.103$_{\pm 0.013}$ | 0.791$_{\pm 0.012}$ | 0.597$_{\pm 0.033}$ | 0.071$_{\pm 0.050}$ | 0.027$_{\pm 0.023}$ |
| EAT+Fed_BVA | 0.838$_{\pm 0.009}$ | 0.715$_{\pm 0.011}$ | **0.357**$_{\pm 0.024}$ | **0.226**$_{\pm 0.006}$ | 0.735$_{\pm 0.020}$ | **0.632**$_{\pm 0.015}$ | 0.164$_{\pm 0.035}$ | 0.106$_{\pm 0.039}$ |

Table 2: Accuracy of Fashion-MNIST under white-box attacks in IID and non-IID settings

| | CIFAR-10 | | | | CIFAR-100 | | | |
|---|---|---|---|---|---|---|---|---|
| | Clean | FGSM | PGD-10 | PGD-20 | Clean | FGSM | PGD-10 | PGD-20 |
| Centralized | **0.903**$_{\pm 0.003}$ | 0.288$_{\pm 0.001}$ | 0.206$_{\pm 0.001}$ | 0.074$_{\pm 0.005}$ | **0.741**$_{\pm 0.003}$ | 0.166$_{\pm 0.012}$ | 0.049$_{\pm 0.004}$ | 0.032$_{\pm 0.003}$ |
| FedAvg | 0.890$_{\pm 0.002}$ | 0.225$_{\pm 0.022}$ | 0.207$_{\pm 0.004}$ | 0.062$_{\pm 0.008}$ | 0.730$_{\pm 0.003}$ | 0.161$_{\pm 0.009}$ | 0.113$_{\pm 0.009}$ | 0.035$_{\pm 0.006}$ |
| FedAvg_AT | 0.890$_{\pm 0.003}$ | 0.280$_{\pm 0.021}$ | 0.295$_{\pm 0.006}$ | 0.099$_{\pm 0.014}$ | 0.707$_{\pm 0.003}$ | 0.162$_{\pm 0.006}$ | 0.064$_{\pm 0.007}$ | 0.048$_{\pm 0.003}$ |
| Fed_Bias | 0.890$_{\pm 0.004}$ | 0.280$_{\pm 0.018}$ | 0.297$_{\pm 0.011}$ | 0.103$_{\pm 0.012}$ | 0.702$_{\pm 0.002}$ | 0.163$_{\pm 0.005}$ | 0.165$_{\pm 0.007}$ | 0.061$_{\pm 0.003}$ |
| Fed_Variance | 0.889$_{\pm 0.001}$ | 0.267$_{\pm 0.014}$ | 0.276$_{\pm 0.006}$ | 0.092$_{\pm 0.009}$ | 0.710$_{\pm 0.007}$ | 0.161$_{\pm 0.005}$ | 0.157$_{\pm 0.010}$ | 0.045$_{\pm 0.016}$ |
| Fed_BVA | 0.889$_{\pm 0.003}$ | 0.286$_{\pm 0.013}$ | 0.301$_{\pm 0.003}$ | 0.104$_{\pm 0.012}$ | 0.709$_{\pm 0.003}$ | 0.163$_{\pm 0.007}$ | 0.165$_{\pm 0.008}$ | 0.062$_{\pm 0.005}$ |
| EAT | 0.833$_{\pm 0.003}$ | 0.596$_{\pm 0.003}$ | 0.667$_{\pm 0.007}$ | 0.561$_{\pm 0.002}$ | 0.661$_{\pm 0.001}$ | 0.267$_{\pm 0.002}$ | 0.206$_{\pm 0.002}$ | 0.188$_{\pm 0.001}$ |
| EAT+Fed_BVA | 0.833$_{\pm 0.003}$ | **0.598**$_{\pm 0.002}$ | **0.668**$_{\pm 0.001}$ | **0.564**$_{\pm 0.003}$ | 0.657$_{\pm 0.002}$ | **0.272**$_{\pm 0.003}$ | **0.332**$_{\pm 0.003}$ | **0.211**$_{\pm 0.002}$ |

Table 3: Accuracy of CIFAR-10 and CIFAR-100 under white-box attacks

2019), defense (or robust model training) (Madry et al., 2018; Carlini et al., 2019; Tramèr et al., 2018) and interpretable adversarial robustness (Schmidt et al., 2018; Tsipras et al., 2018).

**Federated Learning:** Federated learning with preserved privacy (Konečnỳ et al., 2016; McMahan et al., 2017; Hard et al., 2018) and knowledge distillation (Chang et al., 2019; Jeong et al., 2018) has become prevalent in recent years. Meanwhile, the vulnerability of federated learning to backdoor attacks has also been explored by (Bagdasaryan et al., 2018; Bhagoji et al., 2019; Xie et al., 2019). Following their work, multiple robust federated learning models (Fang et al., 2019; Pillutla et al., 2019; Portnoy & Hendler, 2020; Mostafa, 2019) are also proposed and studied. In this paper, we studied the federated learning's adversarial vulnerability after model deployment from the perspective of bias-variance analysis. This is in sharp contrast to the existing work that focused on the model robustness against the Byzantine failures.

**Bias-Variance Decomposition:** Bias-variance decomposition (Geman et al., 1992) was originally introduced to analyze the generalization error of a learning algorithm. Then, a generalized bias-variance decomposition (Domingos, 2000; Valentini & Dietterich, 2004) was studied in the classification setting which enabled flexible loss functions (e.g., squared loss, zero-one loss). More recently, bias-variance trade-off was experimentally evaluated on modern neural network models (Neal et al., 2018; Belkin et al., 2019; Yang et al., 2020).

## 7 CONCLUSION

In this paper, we proposed a novel robust federated learning framework, in which the aggregation incurred loss during the server's aggregation is dissected into a bias part and a variance part. Our approach improves the model robustness through adversarial training by supplying a few bias-variance perturbed samples to the clients via asymmetrical communications. Extensive experiments have been conducted where we evaluated its performance from various aspects on several benchmark data sets. We believe the further exploration of this direction will lead to more findings on the robustness of federated learning.

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
