# OpenReview forum: "Adversarially Robust Federated Learning for Neural Networks"
_ICLR.cc/2021/Conference — Reject_

### Official Review · AnonReviewer3 · 2020-10-25
**Good on the technical side, but the motivation and setup are somewhat unclear**

**Rating:** 4
**Confidence:** 4

**Review:**

Paper summary:

The paper studies adversarial robustness in the context of federated learning. The authors provide an algorithm for adversarial training that generates adversarial examples on a trusted public dataset and iteratively sends them to the clients, so that they can perform learning on the adversarial examples as well. Notably, the adversarial examples are created by inspecting both the bias and the variance of the current set of models. The method is tested empirically on a wide range of datasets and compared to adversarial training using the local clients' data.

##########################################################################

Pros:

- Both adversarial robustness and federated learning are important concepts in modern machine learning research and are of interest to a wide audience.
- In general, I find the idea of generating adversarial examples based on both the model bias and variance (instead of the bias only) quite interesting.
- The experimental evaluation is done on a wide variety of datasets and sufficient details (and code) are provided, so that reproducibility is ensured.

##########################################################################

Cons:

My main concerns are related to the way that the work is motivated and positioned with respect to prior work. I also find a partial mismatch between what the paper claims and what the experiments seem to suggest.

1. Initially, robustness in the context of federated learning is motivated as an important problem because federated learning could "lead to a degenerate model, which collapses in performance when the corrupted updates are uploaded and aggregated on the server" (citing the abstract) and because federated learning "is susceptible to clients' corrupted updates (e.g. system failures, adversarial manipulations, etc.)" (citing the introduction). I completely agree with these points and therefore I find the problem of robust federated learning an important one.

However, these type of problems have to do with problems in the training procedure (for example, poisoning or backdoor attacks) that can lead to a poor model being learned. Instead, the problem addressed in the paper later on is the one of adversarial examples. As is well known, this is a problem that occurs at test time and is present even in models that are trained properly on clean data. Therefore, I see a mismatch between what problem is motivated and what problem is solved in the paper.

2. While I do think that considering bias and variance in the context of adversarial training is an interesting problem, the paper does not provide any motivation for such analysis. The bias-variance trade-off is a classic way to study the generalization of machine learning models. However, the problem of adversarial examples is present even in model that generalize well to (clean) test data. Why should considering the variance help in this case as well? Has the bias-variance trade-off been studied previously in the context of adversarial robustness (for models training a single dataset)? I believe that these are important points that should be discussed in the paper.

3. I find some of the notation and definitions in the paper slightly confusing. In particular, Definition 2.2 features a bunch of expectations, while the distributions that these expectations correspond to are not defined. In addition, while $y_* (x)$ is a function, the right-hand side of its definition does not seem to involve a dependence on x. While I believe that I understand what is meant in these equations, I think that the preliminaries section should rewritten in a slightly more precise manner.

A similar problem is that the bias and variance notions in equation (3) are not really defined. Definition 2.3 introduces the bias and variance of a single model. Is the bias of K models the average bias of the K models, or the bias of the average of the K models?

4. While the experiments are well-designed and enough detail is given for them to be reproducible, very often the baseline of (Tramer et al 2018) or a mixture of Tramer et al and FED_BVA outperforms the proposed method (FED_BVA). Moreover, it's hard to deduce from the experiments which method performs the best, since this changes between the datasets and also no error bars corresponding to multiple repeats of the experiments are given. In this sense, I think more analysis should be provided about when each of the algorithms should be preferred and in general how should the results be interpreted.

##########################################################################

Review summary:

The paper studies an important problem and provides a wide set of experiments for testing various algorithms for adversarial training. I believe that an interesting set of ideas is present in the paper, but currently they are not systematized and motivated properly. It is therefore hard to evaluate the theoretical and experimental insights that are presented. This is why I do not recommend acceptance for the paper in its current shape.

---

> ### Author Response · Authors · 2020-11-20
> **Feedback for reviewer 4**
>
> We appreciate the reviewers' careful assessment and constructive comments. In the following, we address your review in the format of Q&A.
>
>
> **Q**: The unclear problem motivation? **A**: The problem we studied is adversarial robustness. We target at training a federated learning model that is robust towards adversarial attacks after the model has been deployed. Thanks for pointing out the ambiguity in the abstract and introduction, we have modified the related parts in the manuscript to clarify this motivation.
>
> **Q**: Connections with poisoning attacks and backdoor attacks? **A**: We believe that our work does share some similarities with the reviewer mentioned training scenarios, such as poisoning attacks and backdoor attacks. For example, the learning process is clean (no malicious behaviors are observed during training), but some intentionally generated adversarial poisoning data is mixed with clean data. Meanwhile, recent work [1] has shown that even adversarial trained models can be robust towards the backdoor attacks.
>
> [1] Ezekiel Soremekun, et al., AEGIS: Exposing Backdoors in Robust Machine Learning Models, Arxiv 2020.
>
> **Q**: Why is variance important? Has BVD been studied in previous work regarding adversarial robustness? **A**: The generalization error of bias-variance decomposition takes the data set distribution into consideration. In federated learning, since data examples are distributed among clients when we analyze the aggregation error, it is very natural to think of the error incurred by both the bias and variance because generalization error’s computation needs to take an expectation over the data set distribution which matches perfectly with the federated learning setting. To our best knowledge, we believe our work is the first one that discusses adversarial robustness under federated learning using bias-variance decomposition.
>
>
> **Q**: $y_*(x)$ has no dependence on $x$? **A**: Indeed, there is no $x$ involved on the right-hand side of $y_*(x)$. That is because the optimal prediction $y_*$ is defined on every example $(x,t)$ where $x$ is input feature and $t$ is the non-deterministic class label of this example. Thus we use $y_*(x)$, instead of just $y_*$, to denote the optimal prediction over one specific example $(x,t)$.
>
>
> **Q**: Definition of Equation (3)? **A**: The formula of bias and variance depends on the loss term being selected. Therefore, we have to give a general definition at its first appearance and give specific definitions in Theorem 4.1 for cross-entropy loss. The bias of the aggregation is defined using the main prediction and optimal prediction. If there is one client model, its bias is undefined because there is no corresponding main prediction. Therefore, for cross-entropy loss, the bias of the K models is equivalent to the average of K models’ losses.
>
> **Q**: Error bars? And how to interpret the results especially for EAT baselines? **A**: We are able to rerun the experiments with multiple runs using different random initializations. The new results with statistical error bars have been released on the revised manuscript. Regarding the interpretation of the results, we separate the table between the baselines with/without local adversarial training (AT) with a horizontal line in Table 1-3. The takeaway is that without local AT, using a bias-variance based robust learning framework will almost always outperform other baselines for defending FGSM and PGD attacks. When local AT is allowed (e.g., the client device has powerful computation ability), using bias-variance robust learning with local AT will mostly have better robustness. We have added this interpretation into the updated manuscript.

---

> > ### Comment · AnonReviewer3 · 2020-11-22
> > **Thank you for the clarifications.**
> >
> > Thank you for the clarifications. With error bars and details about the experimental setup reported, I believe that the experimental section is quite strong. However, I do feel that the motivation and definitions sections need quite some updating, despite the new edits. Therefore, I would recommend resubmitting an updated version of this paper to another venue.

---

### Official Review · AnonReviewer1 · 2020-10-27
**Interesting paper but falls short**

**Rating:** 5
**Confidence:** 3

**Review:**

The authors propose a robust federated learning algorithm, where they assume that all samples are iid, and $n_s$ clean samples are available at the server side. The authors then go on to optimize a loss function that optimizes the aggregate loss and propose some new algorithms with experimental results. While overall the paper is interesting, there are several shortcomings in the execution as discussed below that the authors can address to improve the paper.

**C1.** Can you please clarify how your Definition 2.3 results in a *decomposition* in the sense of (Pedro 2000), especially given that you remark that this is different compared to the existing definition? Existence of such decomposition is not obvious at all.

**C2.** In bias-variance decomposition, the weights of different components is fixed. In your case $\lambda$ is left as a hyperparameter. Does this mean that you are no after a decomposition and rather would like to balance some notion of bias with a notion of variance. If that is the case, then you need to rewrite the paper to reflect that.

**C3.** In the beginning of Section 3, the authors write *A typical framework (Kairouz et al., 2019) of privacy-preserving federated learning can be summarized as follows*. Can you please comment on how this framework is privacy-preserving?

**C4.** The authors assume that the data is IID, and also some shared and non-adversarial data is available at the central server. While it is okay to assume IID data for an initial theoretical analysis, the latter assumption is not justified especially when studying robustness. Can you please explain how such shared dataset is created?

**C5.** Can you please add statistical error bars to all tables and figures?

**C6.** What happens if $n_s$ is small or even zero? The baselines you are comparing against do not assume the existence of clean data at the server side.

**C7.** What happens if adversary's power is arbitrary? I don't see any reason why a single adversary cannot completely destroy the model performance for all devices if they can do whatever they want with their updates. There is no explicit mechanism to defend against that in this paper.

**C8.** From Figure 5, it seems that the baseline EAT achieves a better tradeoff point between performance and robustness compared to all of the proposed algorithms. Can you please explain?

**C9.** Can you please do a more thorough study of the *tradeoffs curves between robustness and performance* similar to Figure 5? It would be good to see how the different methods compare in terms of the tradeoff between robustness and performance (and repeat that for different notions of robustness). The current tables reporting the results are inconclusive.

---

> ### Author Response · Authors · 2020-11-20
> **Feedback for reviewer 3**
>
> We appreciate the reviewers' careful assessment and constructive comments. In the following, we address your review in the format of Q&A.
>
>
> **Q**: How to get the decomposition? **A**: We inherit the definition of bias and variance directly from [1] where they prove (in Theorem 1) that the decomposition format of bias+variance+noise (with corresponding weight) will stand for any real-valued function regardless if the loss is symmetric or not.
>
> [1] Pedro Domingos, A Unified Bias-Variance Decomposition and its Applications. ICML 2000.
>
> **Q**: Why is there $\lambda$? **A**: Unlike the binary classification problem, the tradeoff portion between the bias and the variance doesn’t have a closed-form solution for the multi-class setting. we need to introduce a hyper-parameter lambda to balance the trade-off. Empirically, we set lambda to 0.01 in all experiments. We have added the details to reflect this in the updated manuscript.
>
> **Q**: How is privacy preserved? **A**: In our Theorem 4.1, it has been proved that the aggregated bias and variance can be formulated as the addition of each client’s gradients when the loss of the neural network is cross-entropy. Therefore, it can use additive homomorphic aggregation to secure privacy because bias as well as variance are separable in terms of each on-device gradient. To ensure that, one change is needed: the server sends the clean shared data to each client, then, all gradient related calculations (of bias and variance) are performed on clients. The server just performs secure aggregations (model parameter average, bias gradients average, variance gradients average) in the same manner as FedAvg. It would be slightly different from the current algorithm where the bias and variance are calculated on the server for the sake of saving on-device computational cost. Please check our updated manuscript for the details of the new algorithm.
>
> **Q**: How to generate shared data? **A**: In the experiments, we have held out $n_s$ examples from the training set as the shared data. There is no overlap between the training data, hold out shared data, and testing data.
>
> **Q**: Statistical error bars? **A**: The update-to-date results with error bars have been reported in the revised manuscript. Please kindly check.
>
> **Q**: Fair comparison?  **A**: In the ablation study, we have included the case where $n_s = 0$. Also, we **do** have included $n_s$ shared examples for **all** comparison baselines to guarantee a fair comparison. It is also reflected in our code implementation. Please kindly verify this in sub-section 5.1 and our code submission.
>
> **Q**: Why the adversaries can not destroy the model performance completely? **A**: In our proposed work, we study the adversarial robustness of federated learning, where the adversarial poisoned samples are generated and mixed into clear data during training. The eventually deployed model after training will be robust against adversarial examples in the testing stage. This is an orthogonal research direction that is different from the setting where some clients could have a totally corrupted model update and manipulate its training set arbitrarily.
>
> **Q**: Is EAT the best tradeoff model in the plot of Figure 5? **A**: In Figure 5, the ideal training framework would be the one that has high robustness without sacrificing too much performance. Therefore, the better framework would be the one on the upper right position. In this plot, Fed_BVA and Fed_Bias achieve a better tradeoff.
>
> **Q**: Why not use more robustness notions? **A**: we follow the setting of [2] to use accuracy under adversarial attack as the robustness evaluation metric. This is straightforward to understand and easy to compute. There are other possible robustness evaluation metrics (worst case loss, etc.) as mentioned in [3] which are usually intractable to be exactly computed and also impractical to be estimated in real scenarios due to its high computational cost.
>
> [2] Yisen Wang, et al., On the Convergence and Robustness of Adversarial Training. ICML 2019
> [3] Carlini et al., On Evaluating Adversarial Robustness. Arxiv 2019

---

> > ### Comment · AnonReviewer1 · 2020-11-22
> > **Insufficient clarifications**
> >
> > Thanks for the clarifications! I find my concerns still remaining and tend to vote to reject the paper.
> >
> > [bias-variance decomposition] I am not convinced that you are doing a formal bias-variance decomposition. You are rather promoting a variance reduction via regularization (see https://proceedings.neurips.cc/paper/2017/hash/5a142a55461d5fef016acfb927fee0bd-Abstract.html)
> >
> > [shared data] While I understand how you generated shared data in your simulations, I still don't see how such shared data will be generated in real world federated learning settings.
> >
> > [fair comparisons] Your baselines are not designed to use shared data with certain characteristics, making the comparison unfair.
> >
> > [robustness notion] A good notion is not one that makes computations easy; it is one that can happen in practice and one would need to defend against in real world.

---

> > > ### Author Response · Authors · 2020-11-23
> > > **follow up**
> > >
> > > Thanks for the response!
> > >
> > > [bias-variance decomposition] We had read the paper, and feel like the given paper in the reference is slightly related. But since it is solving a totally different problem, we respectively disagree with your opinion.
> > >
> > > [shared data] We have given the example in Section 2.1, in a real application, this shared data can use the publically accessible data set.
> > >
> > > [fair comparison] Would you please be specific about the "certain characteristics" refer to? We'd love to hear your suggestion in detail and make the experiments more solid.
> > >
> > > [robustness notion] The robustness notion we used agrees with the ones used in most recently adversarially training papers as we pointed out in the reference. If the reviewer has alternative suggestions, would you please be specific and enlighten us on what is your suggested robustness metric? We'd love to adopt your idea if it suits with our problem's evalution.

---

### Official Review · AnonReviewer2 · 2020-10-28
**A good submission with some weaknesses**

**Rating:** 6
**Confidence:** 3

**Review:**

1/ Summary of contributions

This paper investigates the problem of training a neural network in a federated fashion so that the final model will be resilient to adversarial perturbation attacks. Assuming that a shared public dataset exists, a novel method called FedBVA (Bias-Variance attacks) is introduced: it consists in crafting new adversarial inputs on the server by minimising a custom loss function at each aggregation round, and sharing these new inputs to all clients.
Experimental results on standard image datasets (MNIST, fashion-MNIST, CIFAR) split in iid and non-iid (by class) settings show that the proposed approach yields a more robust model than standard FedAvg in both cases, at the expense of a slight drop in accuracy.

2/ Acceptance decision

Owing to its well conducted experimental section and the novelty of the problem tackled, I would tend to accept this paper, even if it has some weaknesses in the algorithm section.

3/ Supporting arguments

A/ Experimental results. The authors perform a thorough investigation of the proposed approach, comparing it with all reasonable baselines: local adversarial training (EAT), adversarial training done at the server level but with a more standard loss function (FedAvg_AT), individual terms of the custom loss function and the combination of their approach and EAT. The breadth of the investigation (baselines, 3 datasets, 2 attacks) and the number of ablation studies provided in the appendix is quite convincing.

B/ Novelty. I think this is the first time the problem of training an adversarially robust model in the Fl setting is investigated. Even if the clients are not assumed malicious here, it might open new perspectives for treating backdoor or byzantine attacks.

C/ Weakness in the algorithm section. FedBVA relies on a bias-variance decomposition of the loss (cross-entropy and MSE), which allegedly builds upon the formalism of Domingos ’00 and Valentini & Dietterich ’04. However, I have some doubts on the justification of this approach, for the following reasons:
- The authors slightly change the definition of the bias and variance, and in the cross-entropy case they are not equivalent due to the asymmetry of the cross-entropy loss in its both arguments. This slight change is instrumental in permitting a nice expression of these terms in Theorem 4, but not justified beyond the fact that both reduce to the same in the case of a symmetric loss (such as MSE).
- Further, none of the related works cited by the authors explicit the existence of this bias-variance decomposition in the case of a cross-entropy loss, which makes it difficult to check if the decomposition holds in this case.
- Last, but not least, the authors never explicit such a decomposition: they state it as a combination Bias + \lambda * Variance, but do not provide either a closed-form expression for \lambda nor a list of numerical values used in the experiments.

4/ Additional comments

1. In order to understand how significant the robust FL results are, it would be relevant to report the performance under attack of a model trained in a pooled fashion, i.e. with all data in 1 client.
2. From what I understand, this paper does not assume that clients are having a malicious behaviour, which is typically what previous works in FL tried to bring robustness to. It would be nice to comment on this difference, and it would help to clarify the problem tackled by this paper.
3. Could you provide more comments on the subtle differences in Figure 1? It is difficult to see them and it would be profitable in order to better understand the contributions of both loss terms.
4. Interestingly, the proposed FedBVA bears some similarity with distillation-based approaches, such as (Jeong et al. 2018 https://arxiv.org/abs/1811.11479) or (Chang et al. 2019, https://arxiv.org/abs/1912.11279). I would recommend citing these related works for reference.
5. The FedBVA approach needs to have access to individual updates sent by clients. In some FL settings, this is an unacceptable breach of privacy, and secure aggregation is used to ensure that the server only sees the mean update. In this case, the proposed FedAvg_AT is a better compromise than FedBVA.
6. For Figure 2, were experiments done on a trained or untrained network? Are the bias and variance terms computed on training or testing samples?
7. In Figure 4 and Figure 5, for which attack are is the model performance represented?
8. In the second paragraph of Section 5.1, what are the defence models? Is it the model used for FL training?

Some typos:

9. In Appendix A.2, in the derivation of the bias, the terms $\log(1-\log t)$ should be changed as $\log(1-t)$
10. The citation (Pedro, 2000) should be changed as (Domingos, 2000)
11. In Figure 2, the legend has 'variancee' instead of "variance"
12. In the first paragraph of section 5.1, " the baselines (5) and (6) " -> " (6) and (7) "

---

> ### Author Response · Authors · 2020-11-20
> **Feedback for reviewer 2**
>
> We appreciate the reviewers' careful assessment and constructive comments. In the following, we address your review in the format of Q&A.
>
> **Q**: Justification of the bias-variance decomposition on the asymmetric cross-entropy loss. **A**: To the best of our knowledge, we are the first work that uses the bias-variance decomposition of [1] with cross-entropy loss. Regarding the decomposition on asymmetric loss, we know that Theorem 1 of [1] proves that the decomposition format of bias+variance+noise (with corresponding weight) will stand for any real-valued loss function in the two-class problem regardless if the loss is symmetric or not. For the multi-class case, we inherit their definition of bias and variance directly. However, the $\lambda$ coefficient in front of variance doesn’t have a closed-form solution for multi-class scenario, so we treat it as a hyper-parameter to tune. Empirically, we have tried $\lambda$ in a list of values [-10, -1, -0.1, -0.01, 0.01, 0.1, 1, 10], and found $\lambda$ has a generally better performance when it is set to 0.01 in all the experiments. We have added these details in the updated manuscript.
>
> [1] Pedro Domingos, A Unified Bias-Variance Decomposition and its Applications. ICML 2000.
>
> **Q**: Add experiment of performing AT with one client? **A**: We have added the experiment with only one client (namely, the same fashion as the centralized model training). The update-to-date results have been reported in the revised manuscript. Please kindly check.
>
> **Q**: What if the client has malicious behavior? **A**: We studied the robustness of FL in an orthogonal direction, namely the adversarial robustness which focuses on the robustness of the final deployed model. This is different from the traditional robustness that FL focuses on, such as Byzantine failure. Our basic assumption is that the learning process is clean (no malicious behaviors are observed during training), however, the intentionally generated adversarial poisoning data is mixed with clean data during training. The eventually trained model being deployed on the devices will be robust against future adversarial attacks. A very good application would be federated hospital systems, where each hospital (client) could contribute a smaller set of clean MRI images or ECG signals during training, and ends up deploying a robust model against potential future adversarial perturbed medical images or signals. We have pointed out this difference and add it to the updated manuscript.
>
> **Q**: More clarification in Figure 1?  **A**: The plot in Fig.1 is designed to visualize the importance of variance. Although the formula of variance (needs model output and main prediction) is very different from the formula of bias (needs model output and target label), their eventual gradient visualizations are highly similar but not the same. This phenomenon has also been verified in the results table, which shows that variance is also very important towards adversarial training.
>
> **Q**: Additional references? **A**: Thanks for pointing them out, they are indeed related, and we have cited them in the related work.
>
> **Q**: How to guarantee a secure aggregation with privacy being preserved? **A**: In our Theorem 4.1, it has been proved that the aggregated bias and variance can be formulated as the sum of each client’s gradients when the loss of the neural network is cross-entropy. Therefore, it can use additive homomorphic aggregation to secure privacy because bias as well as variance are separable in terms of each on-device gradient. To ensure that, one change is needed: the server sends the clean shared data to each client, then, all gradient related calculations (of bias and variance) are performed on clients. The server just performs secure aggregations (model parameter average, bias gradients average, variance gradients average) in the same manner as FedAvg. It would be slightly different from the current algorithm where the bias and variance are calculated on the server for the sake of saving on-device computational cost.
>
> **Q**: Experimental setting of Figure 2? **A**: The plots are generated using a trained network, and the bias and variance are calculated on training samples.  We have added these details in the manuscript, thanks for pointing it out.
>
> **Q**: Experimental setting of Figure 4, 5? **A**: We plot both using PGD-20 attack, we have added the details in the manuscript, thanks for pointing it out.
>
> **Q**: What is the defence model? **A**: We name the model used for FL as the defense model which will defend against adversarial attacks.
>
> **Typos**: All typos have been fixed, thanks for the corrections.

---

### Official Review · AnonReviewer4 · 2020-10-29

**Rating:** 4
**Confidence:** 4

**Review:**

This paper proposes FedBVA for robust federated learning.  FedBVA first generates adversarial examples at the server-side, where these adversarial examples are those on which the global model incurs large losses.  Here, the authors choose them by designing a loss function: the sum of average loss (bias) and estimated variance.  The idea seems interesting, but I have a few doubts about its validity and the correctness of the experimental results.

Concerns:
- Secure aggregation:  The proposed scheme seems to require access to individual models, implying that it cannot be used together with secure aggregation.  This doesn't seem fixable as the attack algorithm requires access to individual predictions or individual gradients.  (See the variance gradient expression in p5.).  This somehow contradicts the introduction's claim ``our work bridges this gap ... error incurred during secure aggregation ...".  I might be missing something here, so please correct me if I am wrong.  Note that client data is not anymore private if the server has access to individual local models.

- Experimental results:  The ablation study is somewhat incomplete.  What about the performance of EAT + Fed_Bias or EAT + Fed_Variance?  Also, the experimental results need further justification.  Are those numbers based on a single run for each configuration?  Note that FGSM and PGD themselves are highly sensitive to random initializations.  Moreover, non-IID cases also require multiple runs to get a reliable performance estimation.  The performance of EAT significantly decreases as the attack gets stronger in Table 2 and Table 3.  Is EAT using the same set of adversarial examples for adversarial training?  What happens if EAT also uses more iterations when creating adversarial examples when running adversarial training?

- Why is FedAvg robust?: Also, some numbers don't make much sense to me.  For instance, see the first row of Table 1.  Why can't you achieve close-to-zero accuracy with PGD-10 when attacking the vanilla FedAvg model with high epsilon (eps = 0.3)?  What about FGSM?  Isnt' it supposed to bring it down the accuracy close to 10% or so?  For instance, the numbers in Table 1 of [Wang et al, "On the Convergence and Robustness of Adversarial Training"] show that PGD-10 can achieve 0% accuracy with eps = 0.3.  Unless the model trained with standard FedAvg is inherently robust against adversarial examples for some reason, which I doubt, I cannot understand these results.

- Writing: It was tough to read the paper, especially the bias-variance trade-off part.  [Domingos 2000] (not [Pedro 2000]) helped me understand this part, but I believe that the authors can improve the readability of this part.

---

> ### Author Response · Authors · 2020-11-20
> **Feedback for reviewer 1**
>
> We appreciate the reviewers' careful assessment and constructive comments. In the following, we address your review in the format of Q&A.
>
> **Q**: Can the proposed model utilize secure aggregation?
> **A**: One of the common secure aggregation strategies utilized in Federated learning is homomorphic encryption, where the additive model aggregation can securely preserve the privacy of the gradients and model parameters [1]. In our Theorem 4.1, it is proved that the bias and variance can be formulated as the addition of each client’s gradients when the loss is cross-entropy. Therefore, as long as the gradients of bias and variance are separable, the additive homomorphic aggregation can be applied to secure privacy. To ensure that, one change is needed: the server sends the clean shared data to each client, then, all gradient related calculations (of bias and variance) are performed on clients. The server just performs secure aggregations (model parameter average, bias gradients average, variance gradients average) in the same manner as FedAvg. It would be different from the current algorithm where the bias and variance are calculated on the server for the sake of saving on-device computational cost. The corresponding revision of the algorithm has been made in the updated manuscript.
>
> [1] Abbas, et al., A survey on homomorphic encryption schemes: Theory and implementation, CSUR 2018.
>
> **Q**: Do we need to add EAT+Fed_Bias and EAT+Fed_Variance?
> **A**: Due to the space limit, we feel reporting EAT + Fed_BVA will be sufficient for local adversarial training baselines since Fed_BVA performs the best in most cases. Adding local adversarial training to our proposed methods will increase the model robustness but will not change the relative performance ranking between Fed_Bias, Fed_Variance, and Fed_BVA.
>
> **Q**: How about multiple run results?
> **A**: We were able to rerun the experiments with multiple random initializations. The update-to-date results have been updated in the revised manuscript.
>
> **Q**: FGSM and PGD random initialization?
> **A**: All experiments are performed under a fair comparison setting (same random initialization, etc.). As far as we know, FGSM has little randomness, and its results are relatively stable. For PGD, people may add random noise [2] before the adversarial generating step, we turned this functionality off for the sake of fair comparison. Therefore, in our experiments, these two attacking mechanisms have the minimum randomness possible.
>
> [2] Aleksander Madry, et al., Towards Deep Learning Models Resistant to Adversarial Attacks, ICLR 2018.
>
> **Q**: Is EAT using the same set of adversarial examples for adversarial training?
> **A**: EAT performs local adversarial training on each device, where the perturbed data is adversarially generated on the device (plus the shared data among all devices for a fair comparison)
>
> **Q**: What happens if EAT also uses more iterations when creating adversarial examples for adversarial training?
> **A**: We have conducted the experiments of using multi-step attack methods (e.g., PGD) to generate local adversarial examples, please see A.5.2 in the appendix for details. It is possible to perform adversarial training with multi-step attack methods. The immediate effect is that multi-step attack related adversarial robustness will increase, which is quite intuitive. However, it is extremely time-consuming and computationally costly for client devices. Based on our learning goal, using a single-step attack would be sufficient to verify the effectiveness of our framework.
>
> **Q**: Why can't you achieve close-to-zero accuracy with PGD-10 on the vanilla FedAvg model?
> **A**: We observe that the adversary robustness has a correlation with the optimizer being utilized. For example, our alation study in Fig. 9 - 11 regarding the varying momentum of the SGD optimizer shows that larger momentum (e.g., 0.9) will result in a more robust model. We also observe that the SGD trained model could have increasing robustness with an increasing number of FL's communications. Meanwhile, we empirically found that Adam trained models are less robust toward FGSM and PGD attacks. It is very easy to verify: if you have one CNN model trained using Adam on the MNIST data set, you can attack it ($\epsilon=0.3$) using PGD-10 with an accuracy of 0% or using FGSM with an accuracy of 8%. Nevertheless, our objective is to train a robust model (namely, we want a higher-accuracy model). Thus, we carefully verified the optimizer and hyper-parameters and selected SGD with a momentum of 0.9 which has been observed to be relatively robust when the model converges. The reviewer can easily verify our claim with the provided code in the attachment.
>
> **Q**: Feel difficult to read some parts of the paper?
> **A**: We have revised the manuscript as the reviewers suggested. For clarity, we added some content to the bias-variance decomposition section and it is updated in the revised manuscript.

---

### Decision · Program_Chairs · 2021-01-07
**Final Decision**

**Decision:**

Reject

**Comment:**

This paper proposes a method called Federated Bias-variance attacks (FedBVA) to generate adversarial examples for federated learning, which can be used to make the model more robust to adversarial attacks.  All the reviewers found the problem and the approach very interesting. Their concerns include the following main points (please see the reviews for more details):
* The decomposition of the bias and variance can be made more rigorous
* The necessity for a shared dataset of adversarial examples makes the application a little limited
* Need fairer experimental baselines
* Vanilla federated learning is not guaranteed to preserve privacy - the authors should edit this claim in the motivation
* Need compatibility with secure aggregation approaches where the central server cannot access local updates

The authors did do a great job of responding to the reviewers' comments. Given the interest in the problem and the novelty of the idea, I think an improved version of the paper would be quite well-received.